# Total-Factor Energy Efficiency and Its Driving Factors in China's Agricultural Sector: An Empirical Analysis of the Regional Differences

Jianxu Liu [1] , Shutong Liu [2], Jiande Cui [1], Xuefei Kang [1], Qing Lin [3], Rossarin Osathanunkul [4,*] and Changrui Dong [1]

1   School of Economics, Shandong University of Finance and Economics, Jinan 250014, China; 20180881@sdufe.edu.cn (J.L.); jiande@mail.sdufe.edu.cn (J.C.); kangxuefei@mail.sdufe.edu.cn (X.K.); 19850480@sdufe.edu.cn (C.D.)
2   Faculty of Economics, Nankai University, Tianjin 300071, China; 1120221250@mail.nankai.edu.cn
3   Yantai Big Data Center, Yantai 264003, China; linqing@shandong.cn
4   Faculty of Economics, Chiang Mai University, Chiang Mai 50200, Thailand
*   Correspondence: rossarin.o@cmu.ac.th

**Abstract:** Improving agricultural energy efficiency is essential in reducing energy consumption and achieving agricultural sustainable development. This paper aims to measure the agricultural total-factor energy efficiency in China rather than the partial-factor energy efficiency while taking full account of regional heterogeneity and to investigate the driving factors of agricultural total-factor energy efficiency. The empirical results showed that the average value of agricultural total-factor energy efficiency is 0.814 in China, and the technological gap ratio is 0.853. The regional difference in agricultural total-factor energy efficiency was quite obvious. Higher agricultural energy inputs are associated with higher agricultural total-factor productivity. The total value of potential agricultural energy savings in 30 provinces of China reached 1704.41 billion tons of standard coal. In terms of the absolute amount of agricultural energy saving, the amount was largest in the low-energy-input area, which was 113.87 million tons of standard coal, accounting for 66.81% of the total potential saving amount. Furthermore, we used the Tobit model to analyze the influencing factors of agricultural total-factor energy efficiency. We found that the proportion of agriculture to GDP has a positive impact on agricultural total-factor energy efficiency, while the per capita income of farmers, fiscal support for agriculture, the illiteracy rate of farmers, agricultural labor input, and agricultural capital stock have a negative impact on agricultural total-factor energy efficiency. Finally, we proposed policy implications in terms of agricultural technological progress, agricultural infrastructure, technical training, etc.

**Keywords:** green agriculture; energy consumption; energy conservation; sustainable development; total-factor energy efficiency





## 1. Introduction

As the largest developing country in the world, China has always attached great importance to the development of agricultural production and made a series of remarkable achievements [1]. The level of agricultural output and productivity has been continuously raised, the mode of agricultural production has been transformed from an emphasis on human and animal power to an emphasis on mechanical operations, and the development of agricultural modernization has gradually advanced from traditional agriculture [2]. However, the high growth of China's agriculture largely relies on the large input of production factors such as fertilizers, pesticides, and energy, which are part of an extensive agricultural development model that mainly relies on energy consumption [3]. This not only leads to great waste of energy resources but also brings about pollution in air, soil, and

groundwater [4–6]. Compared with the industrial sector, the energy utilization efficiency of the agricultural sector is usually low, and the energy consumption of agricultural activities has an important impact on global environmental conditions and climate conditions [7–9]. In comparison with other countries, China's energy inputs and outputs also lag far behind those of developed countries and even some developing countries, which is essentially due to low energy efficiency [10]. With the further advancement of China's agricultural mechanization and modernization, China's agricultural energy consumption will continue to maintain an increasing trend in the future [11]. Some scholars predict that China's agricultural energy consumption will reach 161.61 million tons of standard coal by 2025, which is almost double that of 2016 [12]. Moreover, China has a vast territory, and there are great differences in energy endowment, technology level, and agricultural development status between different regions. As a result, different regions have different demands for agricultural energy, and there are obvious regional differences in energy utilization efficiency [13,14]. Therefore, improving agricultural energy efficiency and reducing agricultural energy consumption is of vital importance to China [15,16]. It is also necessary to accurately calculate the value of agricultural energy efficiency in China from the perspective of regional differences and then study ways to improve agricultural total-factor energy efficiency (ATFEE) to fundamentally alleviate resource constraints and other problems faced in agricultural development and realize the development of agricultural modernization in China. This will not only effectively reduce the cost of agricultural production but also contribute to the green and sustainable development of agriculture and promote high-quality economic development [17,18].

In recent years, a number of researchers have focused on the issue of agricultural energy consumption, and the measurement of agricultural energy efficiency is one of the key issues that scholars have paid attention to. Existing studies on agricultural energy efficiency have mainly focused on the following aspects:

(1) The measurement of agricultural energy efficiency. Early studies mostly adopted the single-factor energy efficiency calculation method. Although this method is simple to operate, it has disadvantages, such as a lack of unified standards and neglect for substitution effects among different input elements, which leads to great differences and deviations in the calculation results. Later studies mostly adopted the total-factor energy efficiency calculation method. Vlontzos et al. and Zhu et al. evaluated the energy efficiency of agricultural sectors in EU member states using the data envelopment analysis (DEA) model [19,20]. Wu et al. [3], using the panel data of 30 provinces of China from 2002 to 2016, measured agricultural energy efficiency. The results showed that energy efficiency continued to improve over the study period. Jiang et al. [21], taking the agricultural production data of 30 provinces of China from 2000 to and environment performance of the agricultural sector during the research period. It was found that the average value of energy and environment performance is 0.1842, and it shows a downward trend.

(2) Analysis of regional differences and convergence of agricultural energy efficiency. Li et al. selected China's provincial panel data from 1997 to 2014 to calculate the mean energy efficiency of the eastern, central, and western regions of China to measure provincial energy efficiency and found that the ATFEE of the eastern region of China was the highest, followed by the central region and the western region [22]. Fei and Lin measured China's ATFEE from 2001 to 2012 and found that China's agricultural energy efficiency was relatively low and showed significant regional differences [23]. Yu et al. [1] discussed the convergence of energy consumption per unit of agricultural output value in 30 provinces of China from 1995 to 2014 and found that there was sigma-convergence and unconditional beta-convergence in agricultural energy consumption during the study period.

(3) The influencing factors of agricultural energy efficiency. Wu et al. analyzed the influence of agricultural industry agglomeration on agricultural energy efficiency by using a spatial econometric model and reached the conclusion that agricultural industry

agglomeration can promote the improvement of agricultural energy efficiency [3]. Liu et al. found that increasing farmers' income can effectively improve agricultural energy efficiency [24]. Jiang et al. found that agricultural mechanization has a negative impact on energy consumption [21]. Pakrooh et al. believed that energy prices are one of the determinants of energy consumption. Therefore, setting the optimal energy price can effectively reduce agricultural energy consumption and improve energy efficiency [25]. In addition, scholars have also conducted a lot of research in predicting agricultural energy consumption [26], exploring the path of agricultural energy conservation and emission reduction [27–29] and studying the relationship between energy consumption and economic development [30,31].

To sum up, it can be said that scholars have conducted in-depth studies on agricultural energy consumption from different perspectives, such as measuring agricultural energy efficiency and studying the relationship between agricultural energy consumption and economic development. They have made positive contributions to reducing agricultural energy input and promoting the green development of agriculture, but there is still room for improvement. In terms of agricultural energy efficiency measurement, most of the existing studies have used traditional technical efficiency evaluation methods to measure China's agricultural energy efficiency to obtain the specific value of agricultural energy efficiency in each province of China. On this basis, regional divisions of the whole country were carried out to calculate the ATFEE of different regions, and then, difference analysis, convergence analysis, and influence factor analysis were carried out. However, technological heterogeneity was not taken into account, and differences in agricultural technology levels between regions were ignored, so the obtained ATFEE values were not accurate enough. In addition, most existing studies divide all provinces into eastern, central, and western regions according to their geographical locations and analyze the differences in agricultural energy efficiency. However, under this classification method, there are great differences in agricultural energy use intensity between provinces within the region [14], so this regional classification method is not appropriate for the study of agricultural energy efficiency.

In order to solve the problem of ignoring the technological differences between groups, Hayami first proposed the idea of the meta-frontier [32]. Based on this, Battese and Rao further improved the evaluation method for technical efficiency and proposed a parametric meta-frontier analysis model [33]. The model not only takes into account the difference in technology levels between different groups but also constructs a common frontier with envelopment property, which establishes a unified standard for efficiency comparisons between different groups. In recent years, the common frontier model has been widely used in many fields [34,35]. Before applying the meta-frontier analysis method to efficiency measurement, the frontier surface should be determined first. At present, there are two ways to determine the frontier surface in the academic circle: one is the nonparametric estimation method, represented by DEA [36,37]; the other is the parameter estimation method, represented by stochastic frontier analysis (SFA) [38,39]. Different from the non-parametric estimation method, the parameter estimation method sets the random error term, divides the error term into inefficiency and random error, and takes the random factors into account in the model, which effectively makes up for the shortcomings of the non-parametric method [39]. Based on the full consideration of regional heterogeneity, this paper aims to estimate agricultural total-factor energy efficiency and energy-saving potential with the aid of the meta-frontier model and to analyze the driving factors of the agricultural total-factor energy efficiency of China. The main contribution of this study can be summarized as follows: First, we used a meta-frontier method that considers regional differences to measure the ATFEE of each province under the premise of reasonably grouping samples, which makes up for the neglect in the existing research for the differences in agricultural technology levels between regions and measures the ATFEE from the perspective of regional differences in China. Second, the influencing factors of ATFEE were analyzed from the perspective of regional differences, and ways to improve ATFEE were studied by taking regional differences into consideration. Finally, when grouping all the

samples, the traditional classification method based on geographical location was no longer used, but the average intensity of the agricultural energy use of each province was used as the grouping basis. This classification method was more scientific in studying the issue of agricultural energy efficiency.

## 2. Research Methods and Data Processing

### 2.1. Model Setting

2.1.1. SFA Model

Referring to the research by Xie et al. [40], we constructed a possible set (T) of agricultural production that reflects the technical level of agricultural production. The input factors of agricultural production were set as labor (L), capital (K), and energy (E), and the only output was the agricultural added value (Y). The production possibility set T is as follows:

$$T = \{(L, K, E, Y) : \text{input elements } (L, K, E) \text{ can bring agricultural added value } Y\} \quad (1)$$

In order to measure ATEEE, we defined the Shephard energy distance function as shown in Equation (2). For any possible output, $D_I \geq 1$, if the input is exactly on the boundary, then $D_I = 1$; if it is within the boundary, then $D_I > 1$, indicating that there is an efficiency loss at this time.

$$D_I(L, K, E, Y) = \sup\left\{\alpha : \left(L, K, \frac{E}{\alpha}, Y\right) \in T\right\} \quad (2)$$

Concerning the choice of the form of the energy distance function, Shephard, Lin, and Wang point out that, compared with other function forms, the translog arithmetic function form is simple and flexible [41]. It can not only consider the interaction between input elements but also effectively reduce errors in model setting. Therefore, the distance function in the form of the transcending logarithm is constructed in this paper as shown in Equation (3).

$$\begin{aligned} LnD_I &= \beta_0 + \beta_Y lny + \sum_a \beta_a lnx_a + \frac{1}{2}\sum_a\sum_b \beta_{ab} lnx_a lnx_b + \frac{1}{2}\beta_{YY}(lny)^2 \\ &+ \sum_a \beta_{aY} lnx_a lny + \beta_T t + \frac{1}{2}\beta_{TT}t^2 + \sum_a \beta_{aT} tlnx_a + \sum_a \beta_{YT} tlny \end{aligned} \quad (3)$$

In the formula, $x_a$ represents the factor input, y represents the output, and t represents the time trend. To satisfy $D_I \geq 1$, the following conditions need to be met:

$$\sum_a \beta_{ab} = \sum_a \beta_{aY} = \sum_a \beta_{aT} = 0, \ \beta_{ab} = \beta_{ba}, \ a \neq b \quad (4)$$

Further, labor (L), capital (K), and energy (E) are put into the agricultural input elements in $x_a$. It is assumed that the energy input is linearly homogeneous, and statistical error, $v_{it}$, is introduced to Equation (3). Let $LnD_{it} = u_{it}$, and the basic form of the stochastic frontier function can be obtained as shown in Equation (5).

$$Ln = -F(\cdot) + u_{it} + v_{it} \quad (5)$$

The input elements are put into the basic form of the stochastic frontier function to obtain the SFA of the agricultural energy factor input, as shown in Equation (6), where i represents the province.

$$\begin{aligned} LnE_{it} &= \beta_0 + \beta_L Lnl_{it} + \beta_K Lnk_{it} + \beta_Y Lny_{it} + \frac{1}{2}\beta_{LL}(Lnl_{it})^2 + \frac{1}{2}\beta_{KK}(Lnk_{it})^2 \\ &+ \frac{1}{2}\beta_{YY}(Lny_{it})^2 \\ &+ \beta_{LK} Lnl_{it} * Lnk_{it} + \beta_{LY} Lnl_{it} * Lny_{it} + \beta_{KY} Lnk_{it} * Lny_{it} + \beta_{LT}t * Lnl_{it} + \beta_{KT}t * Lnk_{it} \\ &+ \beta_{YT}t * Lny_{it} + \beta_T t + \frac{1}{2}\beta_{TT}t^2 + u_{it} + v_{it} \end{aligned} \quad (6)$$

2.1.2. Meta-Frontier Model

Influenced by geographical location, climatic conditions, natural resource endowment, and other factors, there are great differences in agricultural production conditions and development levels between different regions of China. If the traditional evaluation method of technical efficiency is adopted and differences in agricultural technical level between regions are ignored, the measurement results of ATFEE will be seriously affected, and the accuracy and credibility of the results will be reduced. Therefore, in order to measure the ATFEE of different provinces of China and compare the differences in ATFEE between different regions, we apply the meta-frontier model to measure the ATFEE of different provinces of China.

Grouping samples according to a reasonable classification method is the premise of applying the meta-frontier model to measure efficiency. Referring to the research of Ouyang et al. [42], Hassanien et al. [43], and Lin and Du [44], this paper takes the energy consumption per unit of the agricultural output of each province as the classification standard and divides the provinces into high-energy-input areas, medium-energy-input areas, and low-energy-input areas. A frontier surface is formed in each group to represent the input–output characteristics of agricultural production in this group. The energy input cost function of each group is shown in Equation (7).

$$E_{it} = f\left(X_{it}, \beta_{(j)}\right) e^{U_{it(j)} + V_{it(j)}} \equiv e^{X_{it}\beta_{(j)} + U_{it(j)} + V_{it(j)}} \tag{7}$$

In the above formula, j represents group grouping, $E_{it}$ represents the actual agricultural energy input of province i in period t, $f\left(X_{it}, \beta_{(j)}\right)$ represents the frontier of agricultural energy input, $X_{it}$ represents the input of other factors of agricultural production and agricultural increase in province i in period t, and $\beta$ represents the parameter vector of j group to be estimated. $V_{it(j)}$ represents a random error term, subject to $N\left(0, \sigma_v^2\right)$, which is used to measure statistical error, and $U_{it} \geq 0$ represents the technical loss inefficiency term, which measures the gap between the actual agricultural energy input, $E_{it}$, and the minimum agricultural energy input, $f\left(X_{it}, \beta_{(j)}\right)$. This article assumes that $U_{it}$ obeys the gamma distribution and is independent of $V_{it(j)}$.

The meta-frontier cost function of agricultural energy input in China can be expressed as:

$$E_{it}^* \equiv f(X_{it}, \beta^*) = e^{X_{it}\beta^*} \tag{8}$$

where $E_{it}{}^*$ represents the minimum agricultural energy input on the metafrontier of agricultural energy input costs. $\beta^*$ represents the parameter vector to be estimated in the meta-frontier cost function. $f(X_{it}, \beta^*)$ shows the envelope curve of the energy input cost front of each group. Therefore, the constraint $X_{it}\beta^* \leq X_{it}\beta_{(j)}$ must be satisfied.

Further, Equation (7) can be rewritten as:

$$E_{it} = e^{U_{it(j)}} \times \frac{e^{X_{it}\beta_{(j)}}}{e^{X_{it}\beta^*}} \times e^{X_{it}\beta^* + V_{it(j)}} \tag{9}$$

The group-frontier agricultural total-factor energy efficiency (GATFEE$_{it}$) can be expressed by Equation (10), which is the reciprocal of the first term on the right side of Equation (9) of province i in period t.

$$GATFEE_{it} = \frac{e^{X_{it}\beta_{(j)} + V_{it(j)}}}{E_{it}} = e^{-U_{it(j)}} \tag{10}$$

The technological gap ratio (TGR) between the group frontier and the meta-frontier can be expressed by Equation (11), which is the reciprocal of the second term on the right side of Equation (9). TGR$_{it}$ measures the ratio of the minimum energy input on the meta-cost frontier to the minimum energy input on the group-cost frontier in province i

and period t. It can be seen from the definition that the value range of TGR is between zero and one; the closer it is to one, the closer it is to the meta-frontier and the smaller the technological gap is. Conversely, the closer it is to zero, the farther it is from the meta-frontier and the larger the technological gap is.

$$TGR_{it} = \frac{e^{X_{it}\beta^*}}{e^{X_{it}\beta_{(j)}}} \tag{11}$$

The definition of meta-frontier agricultural total-factor energy efficiency MATFEE) is shown in Equation (12).

$$TE_{it}^* = \frac{e^{X_{it}\beta^* + V_{it(j)}}}{E_{it}} \tag{12}$$

It can be seen from Equations (10)–(12) that the MATFEE satisfies the following relationship:

$$MATFEE_{it} = GATFEE_{it} \times TGR_{it} \tag{13}$$

Since the GATFEE and the TGR are between zero and one, the MATFEE is still between zero and one and less than the GATFEE.

Based on the measured MATFEE, the agricultural energy-saving potential (ESP) can be calculated, as shown in Equation (14):

$$ESP_{it} = E_{it}(1 - MATFEE_{it}) \tag{14}$$

*2.2. Variable Selection and Data Processing*

In this paper, the ATFEE of 30 provinces in China from 2005 to 2017 was calculated by selecting the data of 30 provinces (excluding Tibet because of an absence of data). Referring to the research of Blancard and Martin [45], the output and input indexes of agricultural production were selected as follows.

2.2.1. Output Indicators

(1)   Value-added agriculture (**Y**).

In this paper, the added value of the primary industry was selected as the output index. In order to eliminate the influence of price factors, we used the price index to deflate the added value of the primary industry and convert it into a constant price based on 1978.

2.2.2. Input Indicators

(1)   Agricultural labor input (**L**).

Labor input in agriculture is expressed by the number of people employed in primary industry. Data were obtained from the China Statistical Yearbook and the Provincial Statistical Yearbook from 2006 to 2018.

(2)   Agricultural capital stock (**K**).

In this paper, the perpetual inventory method was adopted, the capital stock and depreciation rate in the base period referred to the research by Zong and Liao [46], and the agricultural capital stock data were updated to 2017 for each province. Moreover, the agricultural capital stock data of each province were converted into the constant price, based on 1978 with the agricultural means of production price index from 2005 to 2017. The missing data for the agricultural means of production price index were made up of the national agricultural means of production price index of the same year. The data for the provincial agricultural fixed-asset investment and agricultural means of production price index came from data published by the China Rural Statistical Yearbook, the Provincial Statistical Yearbook, and the website of the National Bureau of Statistics.

(3)    Agricultural energy input (**E**).

Since the amount of energy consumed for agricultural production in each province was not directly available, the conversion coefficient of the China Energy Statistical Yearbook was used in this paper. Energy consumption includes coal, oil, and electricity. All energy data were collected from the China Energy Statistical Yearbook and converted into the standard coal equivalent. To determine provincial agricultural energy for every year (including other raw coal, coal washing, coal washing, coke, gasoline, kerosene, diesel oil, fuel oil, liquefied petroleum gas, natural gas, heat, electricity, and other energy), we converted the actual consumption of said provincial units of standard coal energy consumption energy inputs a year into agricultural production [17]. Considering the impact of land factor input on agricultural production, this paper took the cultivated area of crops as the proxy variable of land factor input in agricultural production and divided the agricultural input–output index by the sown area of crops to eliminate the impact of agricultural land input. In other words, the input and output indexes of agriculture are the input of production factors and agricultural added value per 10,000 hectares. The reason for this is that land input may cause multicollinearity because it is strongly correlated with labor and capital inputs [47]. Also, there is a wide variation in land across provinces, indicating the need for the normalization of output and input variables, especially when estimating a stochastic meta-frontier model. In addition, this paper uses the value-added agriculture, agricultural labor input, agricultural capital stock, and agricultural energy input for logarithmic processing and then centralizes the three variables of agricultural added value, agricultural labor input, and agricultural capital stock after logarithmic processing. Finally, the processed data were put into a translog SFA and the meta-frontier model for parameter estimation. Descriptive statistics of agricultural output and input indicators are shown in Table 1.

**Table 1.** Descriptive statistical analysis of variables.

| Variables | Unit | Average | Std | Min | Max |
|---|---|---|---|---|---|
| Value added | Ten thousand CNY per hectare | 0.5409 | 0.3921 | 0.0218 | 2.622 |
| Energy input | Tons of standard coal per hectare | 0.6292 | 0.5612 | 0.0948 | 3.4464 |
| Labor input | People per hectare | 1.9196 | 0.6794 | 0.5064 | 4.0350 |
| Capital stock | Ten thousand CNY per hectare | 0.6635 | 0.8320 | 0.0939 | 8.7082 |

## 3. Estimation of ATFEE in China

### 3.1. Regional Division Based on Agricultural Energy Use Intensity

The premise of using the meta-frontier model to measure efficiency is to reasonably divide all samples according to a certain classification method.

Referring to Ouyang et al. [42] and comparing the definition of carbon emission intensity, we calculated the agricultural energy use intensity of each province (that is, agricultural energy use intensity = agricultural energy use amount/agricultural output value) to reflect the amount of energy consumed per unit of agricultural output. This paper calculates the average value of agricultural energy use intensity in 30 provinces of China from 2005 to 2017 and classifies them according to the average value. We divided China's 30 provinces into three groups, namely, high-energy-input areas (above 0.2), medium-energy-input areas (0.1–0.2), and low-energy-input areas (below 0.1). Table 2 shows the average value and grouping of agricultural energy use intensity in each province. As can be seen in Table 2, Guangxi Province and Inner Mongolia Province, which are also in the western region, have significant differences in agricultural energy use intensity. The same situation also exists in the eastern region and the central region. It can be seen that the traditional method of dividing regions according to geographical location is not suitable for the study of ATFEE.

**Table 2.** Groups based on energy consumption per unit of agricultural output (unit: 10,000 tons of standard coal/100 million CNY).

| Group | Province |
|---|---|
| High-energy-input area (above 0.2) (consisting of six provinces) | Shanxi (0.2829), Xinjiang (0.2758), Shanghai (0.2076), Inner Mongolia (0.2058), Tianjin (0.2037), Beijing (0.2027) |
| Middle-energy-input area (between 0.1 and 0.2) (consisting of nine provinces) | Chongqing (0.1913), Zhejiang (0.1643), Guizhou (0.1624), Gansu (0.1520), Heilongjiang (0.1371), Hunan (0.1330), Yunnan (0.1153), Hubei (0.1103), Ningxia (0.1088) |
| Low-energy-input area (below 0.1) (consisting of fifteen provinces) | Fujian (0.0918), Hainan (0.0916), Liaoning (0.0906), Jilin (0.0903), Hebei (0.0854), Shaanxi (0.0836), Shandong (0.0818), Guangdong (0.0805), Jiangsu (0.0801), Qinghai (0.075), Jiangxi (0.0698), Henan (0.0691), Anhui (0.0588), Sichuan (0.0571), Guangxi (0.0391) |

*3.2. Estimation Results of SFA*

We used the LEMDEP 9.0 software to estimate the stochastic frontier cost function under different forms, and the results are shown in Table 3. Model (1) estimated the random frontier cost function in logarithmic form without adding dummy variables. Model (2) is a log-form model with two grouping dummy variables added on the basis of model (1). Model (3) is a translog model without dummy variables, and model (4) is a translog model with dummy variables added. In the above four models, it is assumed that the inefficiency term, $U_{it}$, obeys the gamma distribution. Judging from the AIC results obtained with different models, the AIC value obtained by model (4) is the smallest, indicating that the stochastic frontier cost function in translog form with dummy variables can better fit the data and is the optimal model. This also verifies the research conclusion of Lin and Wang [41].

From the perspective of the estimated results of model (4), the estimation results of most of the parameters in the model are more significant. The parameter estimation results of the two regional grouping dummy variables are both positive and significant, which proves once again that the ATFEE in different regions in China is different. Thus, it is necessary to study ATFEE from the perspective of regional differences. The interaction terms of grouping dummy variables and input factors show that high- and medium-energy-input areas negatively regulate capital demand for energy. The possible reason for this is that the energy consumption intensity is high in high-energy-input areas and medium-energy-input areas, and the additional capital is used to purchase agricultural machinery and equipment with low energy consumption and high utilization rates, thus reducing the input of energy elements in agricultural production and negatively regulating the demand for energy by capital.

The agricultural labor input and capital stock per unit of land area are positively correlated with agricultural energy input, while the relationship between agricultural added value and agricultural energy input is an inverted "U" shape. An increase in the labor force per unit of land area will increase the agricultural energy input, possibly because the average level of labor input cannot fully play the role of agricultural machinery. Thus, the increase in labor input will improve the intensity of the use of agricultural machinery and equipment, leading to an increase in energy input. With an increase in the agricultural added value per unit of land area, the input of agricultural energy increases first and then decreases, which is in line with the inverted "U"-shaped relationship of the environmental Kuznets curve. With the increase in agricultural added value, more attention is paid to energy consumption, so the input of agricultural energy will be reduced.

**Table 3.** Parameter estimation results of the SFA model.

| Variables | Model (1) | Model (2) | Model (3) | Model (4) |
|---|---|---|---|---|
| Constant | 0.0778 *** (0.0112) | 0.1048 *** (0.0090) | 0.2120 *** (0.0194) | 0.1057 *** (0.0200) |
| L | −0.0458 (0.0635) | −0.0443 (0.0511) | 0.2604 * (0.1385) | 0.6903 *** (0.1304) |
| K | 0.2301 *** (0.0506) | 0.2701 *** (0.0363) | 1.7050 *** (0.1534) | 0.8057 *** (0.1771) |
| Y | −0.0412 (0.0306) | −0.0259 (0.0279) | −0.8578 *** (0.1241) | −1.0473 *** (0.1209) |
| $L^2$ | | | 2.8418 *** (0.4850) | −0.4448 (0.4377) |
| $K^2$ | | | 3.0276 *** (0.1888) | 0.2223 (0.2661) |
| $Y^2$ | | | −1.8332 ** (0.4160) | −1.1066 *** (0.3832) |
| L*K | | | −0.4129 (0.4858) | 1.2210 *** (0.4177) |
| L*Y | | | 1.1836 (0.7926) | 3.7320 *** (0.7703) |
| K*Y | | | −2.2755 *** (0.7178) | −3.6553 *** (0.6349) |
| L*T | | | −0.0139 (0.0184) | −0.1000 *** (0.0158) |
| K*T | | | −0.1497 *** (0.0181) | −0.0234 (0.0165) |
| Y*T | | | 0.1138 *** (0.0185) | 0.1570 *** (0.0162) |
| T | −0.0016 (0.0016) | −0.0030 *** (0.0009) | −0.0208 *** (0.0049) | 0.0071 * (0.0043) |
| $T^2$ | | | 0.0008 ** (0.0003) | −0.0012 *** (0.0002) |
| High-energy area | | 0.2255 *** (0.0087) | | 0.2192 *** (0.0093) |
| Middle-energy area | | 0.0431 *** (0.0066) | | 0.0530 *** (0.0054) |
| High-energy area *L | | 0.1069 (0.1053) | | 0.1504 (0.1155) |
| High-energy area *K | | −0.1835 *** (0.0408) | | −0.3311 *** (0.0964) |
| High-energy area *Y | | 1.4460 *** (0.1126) | | 1.5021 *** (0.1533) |
| Middle-energy area *L | | −0.0021 (0.0854) | | 0.0630 (0.0885) |
| Middle-energy area *K | | −0.4412 *** (0.0760) | | −0.7688 *** (0.0994) |
| Middle-energy area *Y | | 0.7921 *** (0.1286) | | 1.0420 *** (0.1282) |
| AIC | −829.4 | −1092.3 | −935.5 | −1236.4 |
| θ | 10.0026 *** (1.0783) | 12.0698 *** (2.1054) | 9.8463 *** (1.8520) | 17.3779 *** (2.0366) |
| *p* | 1.2649 *** (0.1318) | 0.8081 *** (0.1810) | 0.7798 *** (0.1888) | 1.2321 *** (0.1563) |

Note: The values in brackets are the standard errors of the estimated coefficients. *, **, and *** represent significance at 10%, 5%, and 1% significance levels, respectively, the same as below.

Based on the above analysis, we used the translog stochastic frontier cost function model to estimate the cost function parameter $(\hat{\beta}_{(j)})$ of the high-energy-input group, the medium-energy-input group, and the low-energy-input group and calculate the $GATFEE_{it}$

of each province using Formula (10). For the meta-frontier cost function model, we used quadratic programming with equality constraints to solve the parameter value, $\beta^*$, as shown in Equation (15).

$$\text{MinL}^{**} \equiv \sum_{t=1}^{T} \sum_{i=1}^{N} \left( X_{it}\beta_{(j)} - X_{it}\beta^* \right)^2$$
$$\text{s.t.}\ \ X_{it}\beta^* \leq X_{it}\beta_{(j)}$$
(15)

### 3.3. Overall Analysis of ATFEE in China

The change in ATFEE in China from 2005 to 2017 is shown in Figure 1. During the whole sample observation period, the MATFEE fluctuated between 0.782 and 0.834 in China, with an average value of 0.814. The energy efficiency was relatively high on the whole, but there was still potential room for improvement. From the perspective of energy conservation, this energy efficiency value shows that the current level of agricultural output can be achieved with current state-of-the-art technology by cutting national agricultural energy input by 18.6%. The mean TGR between the meta-frontier and the group frontier was 0.853, indicating that there was a certain gap in agricultural production technology in different regions. In addition, the TGR can also be used to judge the necessity of regional division. The smaller the value of the TGR, the more necessary and scientific the grouping will be. The results again verify that it is necessary to analyze the ATFEE in China from the perspective of regional differences. The GATFEE fluctuated between 0.935 and 0.967, with an average value of 0.954, indicating that the efficiency difference within the group was small. Compared with the energy efficiency measured using the meta-frontier and the group frontier, we found that there are great differences in the estimated energy efficiency values, which were caused by differences in agricultural technology levels between different regions. The empirical results again prove that it is necessary to use the meta-frontier method to measure ATFEE in China.

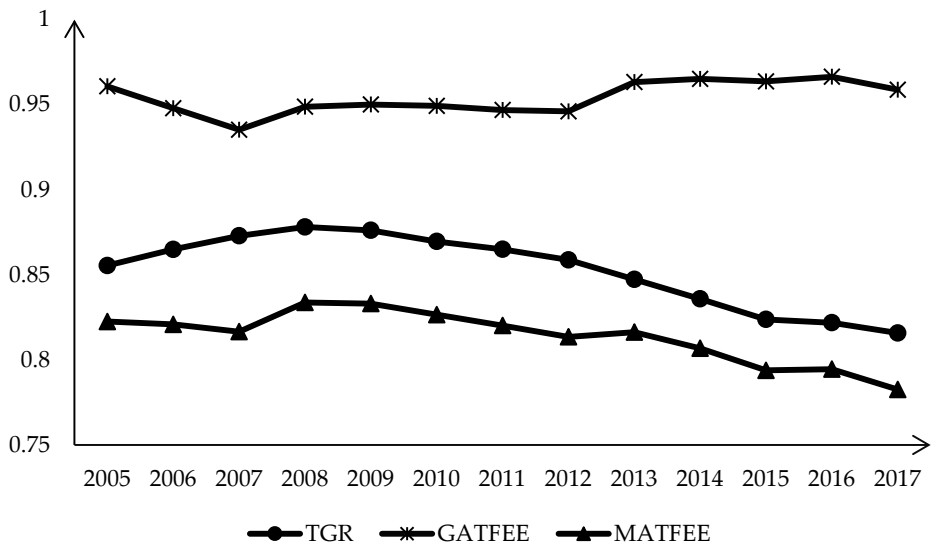

**Figure 1.** Changes in GATFEE, TGR, and MATFEE in China from 2005 to 2017.

In the change trend of efficiency, MATFEE showed a declining trend, the TGR continued to expand, and GATFEE tended to be stable. In the change trend of MATFEE, energy efficiency showed a downward trend from 2005 to 2007; energy efficiency showed an upward trend in 2008; and after 2008, ATFEE showed a slow upward trend. This is because 2005 was the fourth year after China's accession to the World Trade Organization (WTO), and the impact of China's accession to the WTO on China's agricultural production gradually emerged. The fierce international product competition strengthened the volatility of China's agricultural product market, resulting in a decline in ATFEE from 2005 to 2018. In 2008, China formally put forward its agricultural modernization strategy,



which clearly proposed that agricultural production should be transformed from traditional agriculture to modern agriculture. Local governments actively responded to the call of the state and introduced a series of policies and measures, such as subsidies for the purchase of agricultural machinery, to promote agricultural modernization through agricultural mechanization. With the continuous improvement of agricultural mechanization levels in various regions, the utilization efficiency and energy consumption of agricultural machinery were also paid attention to, thus showing a development trend with a slow rise in agricultural energy efficiency. In the change trend of the TGR, the technology gap between the groups narrowed from 2005 to 2008, and then, the technology gap increased year by year. This may be due to the technical gap between regions gradually widening with the improvement of agricultural mechanization. In the change in GATFEE, the average energy efficiency tended to be stable from 2005 to 2017, indicating that the input and utilization of energy elements in agricultural production within each group were basically the same, and the energy efficiency gap within the group was very small.

*3.4. Analysis of ATFEE at the Regional Level in China*

3.4.1. TGR Analysis of Different Groups

Figure 2 shows the TGR trend in three regions of China from 2005 to 2017. The TGR reflects the gap between the optimal technology level in the group and the overall optimal technology level, thus showing the ability of different groups to adopt advanced technology. On the whole, the mean TGR values in the three regions were all less than one, which again indicates that the regional division in this paper is scientific and reasonable. Specifically, the TGR of the high-agricultural-energy-input region was the highest, and its mean value varied from 0.930 to 0.968, indicating that the group frontier of this region is closest to the meta-frontier. The agricultural production technology level of the high-agricultural-energy-input region was the highest. The TGR of the energy-input region was the second highest, and its mean value varied from 0.811 to 0.932, indicating that the group frontier was close to the meta-frontier, and the agricultural technology level of the energy-input region was higher. The TGR of the low-agricultural-energy-input area was the lowest, and its mean value varied from 0.766 to 0.811, indicating that there was a big difference between the group frontier and the meta-frontier in this area, and the agricultural technology level of low agricultural energy input area was the lowest.

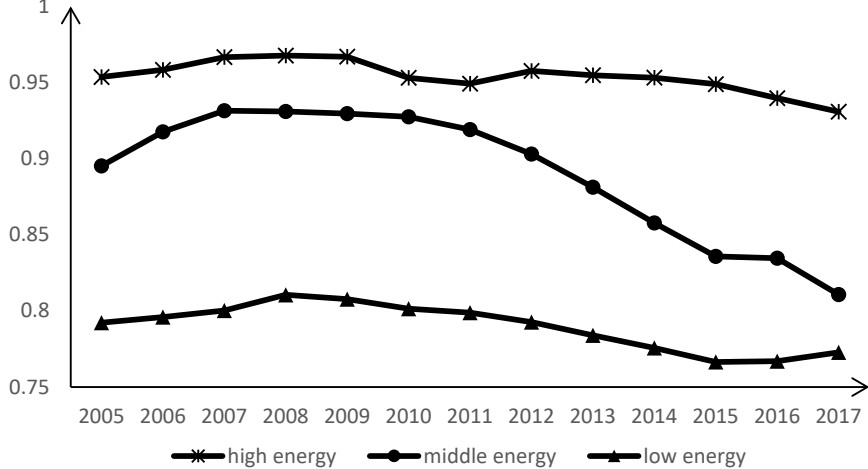

**Figure 2.** Changes in the TGR in different groups from 2005 to 2017.

In the change trend in the TGR between regions, the TGR is relatively stable in high-energy-agricultural-input areas, possibly because high-energy-agricultural-input areas have always adopted advanced agricultural production technology, which represents the most advanced agricultural production technology level to a certain extent. Thus, there is little difference from the meta-frontier. The TGR of the middle-energy-input area in

agriculture showed a decreasing trend to different degrees, and the decreasing speed has obviously accelerated since 2010. This indicates that the distance between the group frontier and the meta-frontier of the energy-input area in agriculture is increasing; that is, the gap between the middle-energy-input area and the most advanced production technology level is becoming larger and larger. The TGR showed a downward trend in the low-energy-input area in agriculture from 2008 to 2015, but it showed an upward trend in other years during the observation period. The widening gap between the technical level in China's agricultural low-energy-input areas and the most advanced technical level may be related to the promotion of agricultural mechanization from 2008 to 2015. The promotion of agricultural mechanization is conducive to improving the level of agricultural technology and promoting the continuous downward movement of the meta-cost frontier, thus increasing the relative distance between the group frontier and the meta-frontier in low-energy-input areas and leading to the widening of the technological gap. The improvement in the TGR means that the group frontier is closer to the meta-frontier, and the gap in agricultural production technology between regions is narrowing continuously.

### 3.4.2. MATFEE Analysis of Different Groups

Figure 3 shows the change in MATFEE in three regions of China from 2005 to 2017. On the whole, MATFEE is the highest in agricultural high-energy-input areas, fluctuating between 0.9 and 0.941, with an average value of 0.918. That is to say, at the average level, high-energy-input areas can gradually approach the meta-frontier technology level by improving their own production technology level, which can increase energy efficiency by 8.2%. The MATFEE of middle-energy-input areas is higher, with an average value of 0.862, which means that the energy efficiency of this region can be improved by 13.8% on average if meta-frontier technology is adopted. The MATFEE of the agricultural low-energy-input area is the lowest, and its value fluctuates between 0.728 and 0.764. It can be seen that there are significant regional differences in ATFEE.

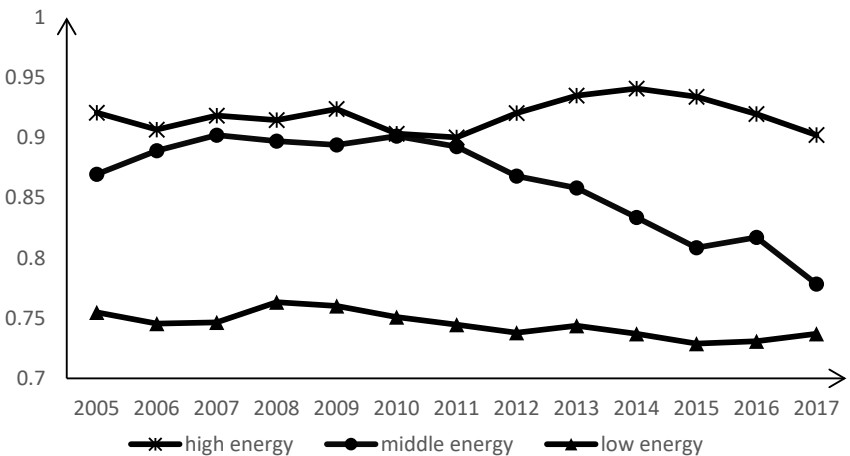

**Figure 3.** Changes in MATFEE in different groups from 2005 to 2017.

In the change trend of ATFEE in each region, the MATFEE of the agricultural high-energy-input region showed an alternately increasing and decreasing trend, but the fluctuation was small, and the overall was relatively stable. The MATFEE of the middle-energy-input area showed an obvious trend of decline, and the decline rate obviously accelerated after 2011. This reflects that middle-energy-input areas in agriculture pay less attention to the problem of agricultural energy consumption. With the continuous advancement of agricultural modernization, the degree of agricultural mechanization has continuously improved. In this process, if we do not pay attention to the utilization efficiency of energy input, it will inevitably lead to a continuous decrease in ATFEE. The energy efficiency of the agricultural low-energy-input area is relatively stable, and there was no significant increase or decrease in the observation period.

### 3.5. Analysis of Potential Saving Amounts in Agricultural Energy in Provinces of China

Based on the ATFEE measured for each province in the meta-frontier, this part of the paper calculates the potential savings in agricultural energy input in the sample observation period for each province, as shown in Table 4. From 2005 to 2017, the total value of potential agricultural energy savings in 30 provinces of China reached 1,704,409 million tons of standard coal. In terms of the absolute amount of agricultural energy saving, the amount was the largest in the low-energy-input area, which was 113,871,900 tons of standard coal, accounting for 66.81% of the total potential saving amount. This is due to the low MATFEE and the large number of provinces included. The potential energy saving amount is 43.5449 million tons of standard coal in the middle-energy-input zone, accounting for 25.55% of the total potential energy savings. Given the high MATFEE and the small number of internal provinces, the absolute value of potential agricultural energy savings in the high-energy-input area is the smallest, which is 13,023,500 tons of standard coal, accounting for 7.64% of the total potential energy savings.

**Table 4.** Agricultural energy-saving potential (10,000 tons of standard coal).

| Group | Province | Energy Savings | Group | Province | Energy Savings |
|---|---|---|---|---|---|
| High-energy-input area (1302.35) | Inner Mongolia | 680.39 | | Shandong | 1700.26 |
| | Shanxi | 217.13 | | Jiangsu | 1391.21 |
| | Xinjiang | 188.07 | | Hebei | 1266.39 |
| | Beijing | 105.25 | | Henan | 1167.80 |
| | Shanghai | 65.97 | | Liaoning | 1072.98 |
| | Tianjin | 44.93 | | Fujian | 849.81 |
| Middle-energy-input area (4354.49) | Hunan | 1070.11 | Low-energy-input area (11,387.19) | Guangdong | 821.74 |
| | Heilongjiang | 983.26 | | Sichuan | 752.20 |
| | Hubei | 703.32 | | Anhui | 507.09 |
| | Chongqing | 502.97 | | Jilin | 499.88 |
| | Zhejiang | 341.23 | | Jiangxi | 353.49 |
| | Yunan | 266.38 | | Shaanxi | 348.27 |
| | Guizhou | 225.50 | | Guangxi | 319.93 |
| | Gansu | 195.97 | | Hainan | 266.80 |
| | Ningxia | 65.75 | | Qinghai | 69.34 |

Figure 4 shows the change in the proportion of potential energy savings in total energy savings in different regions in China from 2005 to 2017. As can be seen from Figure 4, since 2007, the proportion of potential agricultural energy savings in middle-energy-input areas increased year by year, while the proportion in low-energy-input areas decreased year by year, and the proportion in high-energy-input areas has hardly changed.

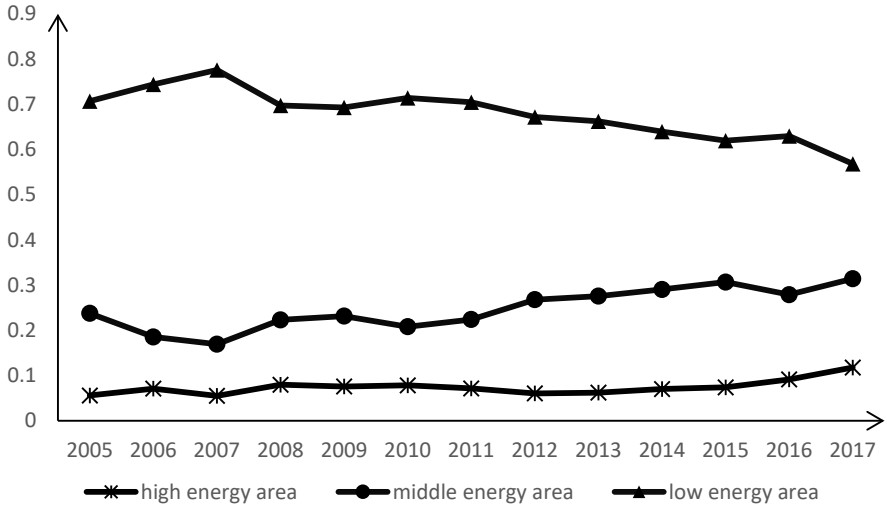

**Figure 4.** The proportion of agricultural energy-saving potential in different groups from 2005 to 2017.

## 4. Analysis of the Influencing Factors of ATFEE

In order to reduce energy input in China's agricultural production and improve energy efficiency effectively, this section is focused on the measurement results regarding ATFEE and analyzes the influencing factors of energy efficiency from the perspective of regional differences. Considering that the ATFEE value measured in this paper is between zero and one, which means that it is a restricted dependent variable, if the traditional linear regression method is used, a negative fitting value may be obtained, leading to biased parameter estimation results [48]. Therefore, the Tobit model was selected in this section to study the influencing factors of ATFEE.

### 4.1. Variable Declaration

Referring to the research of Wu et al. [3] and Fei and Lin [12] and combined with the availability of the data, this paper mainly considers the following influencing factors:

(1)  Rural economic development level

The per capita net income of peasants was used to represent the level of rural economic development in each province. The higher the per capita net income is, the better the rural economic development in the region is, and more attention may be paid to the sustainability of development, thus affecting the use of energy elements in agricultural production. The data were obtained from the China Rural Statistical Yearbook and the Provincial Statistical Yearbook for 2006 to 2018.

(2)  Regional industrial structure

The proportion of agricultural output value in the gross regional product was used to show the industrial development structure of the region, which reflects the importance of agricultural development to the regional economy. Generally speaking, the greater the role of agriculture in regional economic development, the more attention it will receive from various parties. Then, agriculture will receive more financial and policy support, which will have a certain impact on energy efficiency. The data were obtained from the China Statistical Yearbook (2006–2018).

(3)  Financial support

The financial support intensity is expressed by the ratio of the provincial financial support agricultural expenditure to the total fiscal expenditure. The higher the ratio is, the greater the government's support for the progress of agricultural development in the province, which may have an impact on the ATFEE of the province. The data came from the National Bureau of Statistics.

(4)  Rural education level

The illiteracy rate of rural people over the age of 15 in our country indicates the educational level of farmers [49]. The higher the illiteracy rate is, the lower the education level in the region will be, and the labor productivity of farmers will decrease accordingly. Thus, this will affect the utilization rate of agricultural machinery and the promotion and application of advanced technologies in agricultural production and may also have a certain impact on ATFEE. The data came from the China Database on Agriculture, Rural Areas, and Farmers.

(5)  Internal structure of agriculture

The added value of forestry, animal husbandry, and fishery in agriculture accounted for the proportion of the total agricultural added value to show the change in the internal structure of agriculture. Different agricultural sectors have different energy consumption rates, so changes in the internal structure of agriculture may have a certain impact on ATFEE. The data were obtained from the China Rural Statistical Yearbook (2006–2018).

(6) Factor input level

Factor input is expressed by the labor input of the primary industry and agricultural capital stock per hectare of land. The input of agricultural factors in the production process will affect the agricultural production efficiency and thus may have an impact on ATFEE. The data for labor input came from the China Statistical Yearbook and the Provincial Statistical Yearbook for 2006 to 2018, and the data for agricultural capital stock came from the calculation above.

(7) Regional division

Two dummy variables were used to reflect the division of regions, namely, "whether it is an area with high energy input in agriculture" and "whether it is an area with energy input in agriculture". Through the above study, we found that there are great differences in agricultural energy use intensity and utilization efficiency in different regions. Therefore, considering that the influencing factors of ATFEE may also have regional differences, dummy variables of regional grouping were introduced to the model.

*4.2. Analysis of Empirical Results*

Using the STATA version 14 software, the Tobit model was selected to study the influencing factors of ATFEE in China, and the model estimation results are shown in Table 5. On the whole, the influencing factors of MATFEE are quite different from those of GATFEE. From the perspective of the impact of the proportion of fiscal expenditure on agriculture and the illiteracy rate of farmers on agricultural energy efficiency, the two variables have a differentiated impact on MATFEE and GATFEE. The proportion of financial support for agriculture and the rate of rural illiteracy are positively correlated with GATFEE but negatively correlated with MATFEE. The differences in empirical results are precisely caused by differences in agricultural technology levels between regions. Combined with the reality, the impact of the two variables on MATFEE is more in line with the reality of economic and social development, so the MATFEE value calculated based on the meta-frontier is more accurate. Specifically, the impact of each variable on ATFEE is as follows:

(1) An increase in per capita income for farmers leads to a decrease in MATFEE but has no significant impact on GATFEE. The reason may be that increases in rural income levels mainly depend on increases in non-agricultural income, so the proportion of agricultural income in farmers' incomes will decline. Then, farmers will attach less importance to agriculture, which is not conducive to the development of agriculture and will lead to a decline in agricultural energy efficiency. On the other hand, the improvement of farmers' income levels encourages farmers to buy more agricultural machinery and equipment, but for those engaged in agricultural production in the rural labor force, their abilities are generally poor, and they do not have the skills to grasp the operation of agricultural machinery equipment. Thus, this may be the reason agricultural machinery consumes too much energy, which is not conducive to improving the energy efficiency of agriculture.

(2) An increase in the proportion of agriculture in GDP can significantly improve MATFEE but has no significant effect on GATFEE. Therefore, it can be inferred that an increase in the proportion of agriculture in GDP is conducive to narrowing the technological gap between regions. The reason may be that the higher the proportion of agriculture in GDP, the greater the role that agriculture plays in the economy and society. Whether from the perspective of farmers or the government, more attention should be paid to the development of agricultural production, which is beneficial to agriculture. The improvement of the technical level and the improvement of agricultural management have promoted the improvement of agricultural energy efficiency.

(3) The proportion of financial support for agriculture is negatively correlated with MATFEE but positively correlated with GATFEE. That is to say, an increase in the proportion of financial support for agriculture will widen the technological gap between regions. First of all, in reality, more attention is paid to increases in the size

of agricultural machinery and equipment investment, and the problem of energy consumption in equipment is often ignored. Secondly, the maintenance of agricultural machinery and equipment in later periods is also relatively deficient, which leads to a reduction in energy efficiency.

(4) The illiteracy rate of farmers has a negative impact on MATFEE and a positive impact on GATFEE. It can be inferred that an increase in the illiteracy rate will widen the technological gap between regions. This increase in the illiteracy rate indicates that farmers in this area are not well educated, which will lead to a low level of labor productivity and a weak ability to master advanced production technology. Therefore, it is not conducive to the application or promotion of advanced agricultural production technology, which will lead to an increase in the TGR and a decrease in MATFEE.

(5) Change in agricultural industrial structure has no significant influence on MATFEE. We argue that, with the development of the economy and society, the complementarity between forestry, animal husbandry, fishery, and traditional planting has become weaker and weaker in recent years. For example, it is less and less common to use waste generated by aquaculture as fertilizer for crops. Now it is more common to turn generated waste into biogas to meet people's daily needs rather than applying it to agricultural production. Therefore, the impact of agricultural industrial structure on ATFEE is not significant.

(6) The agricultural labor and capital input per unit of land has a negative impact on MATFEE but has no significant impact on GATFEE. It can be seen that the method of promoting agricultural growth only by increasing the factor input is complicated, which is not conducive to improving ATFEE and does not meet the requirements of sustainable development.

**Table 5.** Analysis of the influencing factors of MATFEE and GATFEE.

| Variables | MATFEE | GATFEE |
|---|---|---|
| Constant | 0.8223 *** | 0.9365 *** |
| | (0.0275) | 0.0175 |
| Per capita income of farmers | −0.0187 ** | −0.0010 |
| | (0.0092) | (0.0058) |
| Agriculture in GDP | 0.0015 * | −0.0006 |
| | (0.0008) | (0.0005) |
| Financial support for agriculture | −0.0026 ** | 0.0031 *** |
| | (0.0012) | (0.0007) |
| Illiteracy rate of farmers | −0.0012 * | 0.0011 ** |
| | (0.0007) | (0.0004) |
| Agricultural industrial structure | −0.0004 | −0.000453 ** |
| | (0.0003) | (0.0002) |
| Labor | −0.0104 * | −0.0035 |
| | (0.0054) | (0.0034) |
| Capital | −0.0075 * | −0.0027 |
| | (0.0043) | (0.0027) |
| High-energy area | 0.1787 *** | 0.0252 *** |
| | (0.0106) | (0.0067) |
| Middle-energy area | 0.1198 *** | 0.0164 *** |
| | (0.0084) | (0.0053) |
| Sigma | 0.0635 *** | 0.0405 *** |
| | (0.0022) | (0.14) |

Notes: Standard errors are in parenthesis. ***, ** and * implies significance at the 1%, 5% and 10% levels, respectively.

*4.3. Discussion*

In recent years, several studies have confirmed the assertion that improving energy efficiency promotes sustainable development and energy conservation in agriculture from different perspectives. Fei and Lin [23] argued that, with the continuous advancement of agricultural technology and the vigorous development of agricultural mechanization,

agricultural energy consumption will continue to increase. It is crucial to reduce agricultural energy consumption and carbon emissions by improving agricultural energy efficiency [3]. Therefore, previous studies support exploring the drivers of agricultural energy efficiency. This paper reveals the role of farmer income, agricultural GDP, and farmer illiteracy rates in influencing agricultural energy efficiency, extending the studies of Xu and Lin [50], Cheng et al. [51], Honma and Hu [52], etc. Wu et al. [3] found that agricultural GDP can positively impact agricultural energy efficiency. The Chinese government's policy of promoting green agricultural development also states that agricultural energy efficiency can be improved by increasing agricultural output and reducing agricultural inputs and emissions to realize green agricultural production. Li et al. [53] found that increasing farmers' agricultural incomes can promote agricultural energy efficiency, which also supports our finding that an increase in farmer income can reduce agricultural energy efficiency. Farmers are participants in agricultural production, and the low literacy of farmers directly affects the management of agricultural production. Some scholars have confirmed that managerial inefficiency is the main reason for low agricultural energy efficiency [23,53].

The heterogeneity of agricultural energy efficiency is particularly obvious, which is shared by many scholars with similar views. For example, Wen and Li [54] found that agricultural energy efficiency is significantly higher in China's coastal provinces than inland provinces and lower in large agricultural provinces. This is basically consistent with our finding that high-energy-input provinces have higher agricultural energy efficiency, while low-energy-input provinces have lower agricultural energy efficiency. Yang et al. [55] also found that there are large regional differences in agricultural energy efficiency in China. From this point of view, Chinese provinces should strengthen cooperation in agricultural production as a way to jointly promote the improvement of agricultural energy efficiency. The stochastic meta-frontier model applied in this paper fully takes into account the heterogeneity between regions and can avoid biased estimations of energy efficiency. Many scholars, such as Amsler et al. [56] and Molinos-Senante et al. [57], have confirmed that it is more appropriate to measure efficiency using the stochastic meta-frontier model when heterogeneity exists. From this point of view, the stochastic meta-frontier model should be a good substitute for problems that can be analyzed using the stochastic frontier model if there is heterogeneity in efficiency.

## 5. Conclusions and Policy Recommendations

Many scholars have defined traditional energy efficiency by using the specific values of energy output and energy input. This partial-factor energy efficiency might be misleading and has difficulty measuring potential technical efficiency in energy utilization [23,58,59]. Therefore, this study measured the total-factor energy efficiency of China's agricultural sector and investigated the driving factors of MATFEE. We can conclude that a regional technical gap in agricultural energy efficiency exists, which implies that the meta-frontier model was appropriate in this study. The TGR shows a declining trend year by year, especially in provinces with middle energy inputs. In these medium-energy-input provinces, there are large differences in agricultural total-factor energy efficiency, and the technology gap is widening. Agricultural total-factor energy efficiency was higher in provinces with high energy inputs and lower in provinces with low energy inputs over the time period of our study and has basically not changed much. We can improve agricultural total-factor energy efficiency with influencing factors. Increasing farmers' farm incomes, reducing farmers' illiteracy, and increasing agricultural output are all effective ways to improve total-factor energy efficiency in agriculture. Based on the above research conclusions, this paper puts forward the following policy recommendations:

(1) Given the different situations in agricultural energy utilization in different regions, we should take differentiated agricultural energy-saving measures. For areas with high agricultural energy input, the research, popularization, and application of advanced agricultural technologies should be further promoted. The inputs of energy, labor, and other material elements in agricultural production should be continuously

reduced to form an agricultural development model driven mainly by technological innovation. For areas with middle energy input in agriculture, it is necessary to strengthen exchanges and cooperation with technologically advanced areas, introduce advanced agricultural production technology, improve management capacity, narrow the technological gap with the most technologically advanced areas, and improve ATFEE. For areas with low agricultural energy input, it is necessary to strengthen the study of advanced agricultural production technology to narrow the technological gap and also to strengthen intraregional exchanges, learn from provinces with high ATFEE in the region, narrow the intragroup gap, and improve GATFEE.

(2) We should improve the level of agricultural technology, give play to the driving effect of technological progress on agricultural energy efficiency, and reduce energy input in agricultural production. Under the meta-frontier, China's ATFEE showed a downward trend, which was mainly caused by a widening of the technological gap between the group frontier and the meta-frontier. In the future, it will be necessary to pay attention to the improvement of the agricultural technology level, especially to enhance the energy-factor-oriented technology level, strengthen investment in agricultural science and technology research and development, combine agricultural development goals with resource conservation goals, improve ATFEE, and promote the formation of resource-conserving modern agriculture.

(3) To strengthen technical training and technical guidance for farmers, we should improve the rural labor force's awareness and abilities regarding energy conservation and emission reduction. The energy efficiency of agriculture is directly influenced by agricultural machinery use efficiency. However, the left-behind rural labor force's quality is not high in China, which is the immediate cause of low utilization efficiency regarding agricultural machinery. Given their limited education and professional training, they do not have the professional knowledge necessary for agricultural machinery maintenance, and some of them do not have an awareness of energy saving and emission reduction when using agricultural machinery. The strong popularization of agricultural mechanization is also conducive to raising agricultural incomes and bridging the gap between urban and rural areas. Raising farmers' awareness of energy conservation can effectively improve the rural habitat. In the future, we should strengthen technical training and professional guidance for farmers, popularize advanced agricultural machinery use technology with farmers, improve the comprehensive quality of the rural retained labor force, and improve their awareness and abilities regarding energy conservation and emission reduction.

(4) The government should accelerate the construction of agricultural production infrastructure, eliminate backward agricultural machinery, improve the utilization rate of agricultural machinery, and reduce the unit energy consumption of agricultural machinery. In order to realize the development of agricultural modernization, it is necessary to vigorously develop agricultural mechanization. However, agricultural mechanization is not only reflected in the increase in the amount of agricultural machinery but also should be reflected in the improvement of the utilization efficiency of agricultural machinery. Therefore, when formulating agricultural development plans, the government must strengthen rural infrastructure construction, improve agricultural production conditions, and focus on increasing the amount of agricultural machinery while also increasing the utilization of agricultural machinery. This paper makes some contributions to the measurement and drivers of total-factor energy efficiency in Chinese agriculture, but there are some limitations. First, this paper directly assumes that the inefficiency term is a half-normal distribution without considering other distributions. This might lead to inexact estimations of agricultural energy efficiency. Second, in order to ensure the consistency of data collection, this paper adopts data from before 2017. However, the stochastic frontier model has been improved in many ways, so the stochastic meta-frontier model can also be improved, such as relaxing the independence assumption of the error components [60]. As a

result, in future research, we can use more assumed distributions for the inefficiency term, update the data, and improve the stochastic meta-frontier model.

**Author Contributions:** Conceptualization, J.L. and R.O.; methodology, J.L. and S.L.; software, J.L. and S.L.; validation, J.C. and X.K.; resources, Q.L and C.D.; data curation, J.C. and X.K.; writing—original draft preparation, J.L. and R.O.; writing—review and editing, R.O. and J.L.; visualization, C.D. and Q.L.; supervision, R.O.; project administration, Q.L.; funding acquisition, R.O. All authors have read and agreed to the published version of the manuscript.

**Funding:** This research was financially supported by the Key Soft Science Projects in Shandong Province (Grant No. 2023RZB06050).

**Data Availability Statement:** All data can be obtained by email from the corresponding author.

**Acknowledgments:** This work was supported by the China—ASEAN High-Quality Development Research Center at the Shandong University of Finance and Economics and the "Theoretical Economics Research Innovation Team" of the Youth Innovation Talent Introduction and Education Plan of Colleges and Universities in Shandong Province with financial support, as well as the Faculty of Economics and the Centre of Excellence in Econometrics at Chiang Mai University. We also thank one of the key research bases of Social Science in Shandong Province, the research base of Xi Jinping Economic Thought at the Shandong University of Finance and Economics.

**Conflicts of Interest:** The authors declare no conflict of interest.

## Nomenclature

| | |
|---|---|
| ATFEE | agricultural total-factor energy efficiency |
| SFA | stochastic frontier analysis |
| DEA | data envelopment analysis |
| GATFEE | group-frontier agricultural total-factor energy efficiency |
| TGR | technological gap ratio |
| MATFEE | meta-frontier agricultural total-factor energy efficiency |
| WTO | World Trade Organization |

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
