# Peer review of "Total-Factor Energy Efficiency and Its Driving Factors in China’s Agricultural Sector: An Empirical Analysis of the Regional Differences"

_agronomy, doi:10.3390/agronomy13092332_

Round 1

Reviewer 1 Report

Minor editing check is necessary. For example, in line 276 "That is, ?????? is equal to the product of the ?????? and the ??? " need to be revised.

Minor editing check is necessary.

Author Response

Dear Reviewer,

Thank you so much. We have corrected this point following your comments, and also carefully checked the whole manuscript. We appreciate your suggestions and comments very much.

Reviewer 2 Report

The research article is topically good and interesting that addressed important aspects of total factor energy efficiency for sustainable agricultural development. But there are some flaws in the article that must need to be rectified and further improve the article so that it should be considered for publication after a thorough revision. Article write-up, layout structure and quality of presentation should be improved. English language editing/improvement is also suggested.

Title:

it is suggested to revise the title of the article to make it well-structured and catchier for the potential readers (maybe by a language expert).

Abstract:

The abstract should be revised appropriately after revising the article. Clearly stating the research topic background and research problem, adopted methodology, key outcomes and future direction.

Energy measurement in terms of tons of coal is not suitable. Use some more suitable alternative scales and units.

Introduction:

This part should be in well-structured form, appropriately smaller paragraphs, comprehensively stating the background of the research topic, its importance, relating with existing research literature (merging literature review to this part), knowledge gaps, the contribution of this research in fulfilling those knowledge gaps, main objectives and principle impact/ outcomes of the research.

A separate literature review is inappropriate here, merging properly within the introduction section suggested.

Line 180 to 183: Paragraph consisted of unnecessary text, deletion suggested.

Equations should be in a standard-defined mathematical format.

Energy measurement methods, data collection methods and authentic data sources should be refined and stated properly. Forms and operations of energy inputs/ energy consumption should be defined. Which operations are more energy intensive and how they can be controlled?

Energy input data is from 2005 to 2017 while now it is 2023, what is the value of this data now? it should be considered of current years.

Clearly define the limitations and boundary conditions of the applied models.

The results presentation quality is not appropriate. Both tabular and graphical should be presented in more organized and advanced data presentation forms.

The conclusion is the independent representation of the article, many readers only look at the conclusion to get the basic theme of the research. So the conclusion should be well structured and appropriate format.

Very long paragraphs are discouraged.

Using too many abbreviations in this section is not appropriate, use full forms or define abbreviations as it is recommended by many good journals.

If it is found appropriate, consider suggested improvements and resubmit after thorough revision, improving the quality of the presentation and making the content more precise and appropriate.

Minor English editing is required.

Reviewer 3 Report

Dear authors

This is an interesting paper, assessing important aspects of agricultural production in China. Nevertheless, the following points can be improved:

1.       Justify in a better way the methodological approach for the Land input in your models. If there are previous works with the same approach, please mention them, or if there are not, explain in a more clear way the necessity of this transformation.

2.       Provide policy recommendation for bridging the gap between China’s Urban-rural divide, in order to improve environmental deliverables of people living in rural areas

Reviewer 4 Report

This paper is interesting as according to the authors, aims to measure the agricultural total factor energy efficiency (ATFEE) of 30 provinces in China by using a meta-frontier model and studies its influencing factors in agricultural production data from 2005 to 2017 from the perspective of regional differences. The paper is seeming to be well organized and structured, however, the topic has a clear national, Chinese, interest (with limited international-oriented interest) In my opinion the originality of the study is average, even if the authors are trying to prove the opposite, based mainly on the used methodologies. The paper presents good elements and structure, but some improvements could nevertheless be useful to take interest the readers.

1. The Introduction section is focused on agricultural energy efficiency in China, almost totally. Although there is a strong structure of the section, it is not clear at the end of this section the purpose of the study and missing a brief presentation of the paper structure (there is at the end of the Literature review section).

2. At the end of section 2 are presented the contribution and paper structure, in my opinion, both must transfer at the end of section 1. 

3. There are too many repetitions in the whole manuscript.

4. In my opinion, the manuscript needs a Discussion section and I suggest to the authors to add this section. According to the appropriate practice for Discussion section writing, the authors must compare the findings of their own research to the results of similar national and internationally oriented studies. Additionally, they must go deeper by analyzing the whole results and not stay on the surface. Although in sections 4 and 5 the authors are trying not only to present but to discuss also their results, they stay to the surface by explaining simple the results, and some hypotheses they are doing are surface and not well documented.

5. The first paragraphs of the Conclusions and policy recommendations section (6) contain exclusively repetitions of the results from the previous sections. The policy suggestions paragraphs are more specific and are targeting to the core of the topic. 

6. There are no research limitations and future research.

Round 2

Reviewer 2 Report

It's good to see your interventions for improving the article. It looks good and improved enough to be considered for publication. I recommend it to the editor for a final look and consider it for publication after minor improvements.

Submitting the article for review with the track change function is inappropriate. It should be in plain format and the changes made in the article and answers to the comments should be highlighted and replied in the cover letter with an indication of specific parts where edits were made e.g. section or Line No. etc.

So, please submit it again after finalizing the track changes and other minor corrections of language, grammar and formats for final consideration.

Minor language and grammar corrections are suggested.

Reviewer 4 Report

No comments